# The Expanding Utility of Robotic-Assisted Flap Harvest in Autologous Breast Reconstruction: A Systematic Review

**DOI:** 10.3390/jcm12154951

**Published:** 2023-07-27

**Authors:** Nikita Roy, Christopher J. Alessandro, Taylor J. Ibelli, Arya A. Akhavan, Jake M. Sharaf, David Rabinovitch, Peter W. Henderson, Alice Yao

**Affiliations:** 1Division of Plastic and Reconstructive Surgery, Department of Surgery, Icahn School of Medicine at Mount Sinai, New York, NY 10029, USA; nikita.roy@icahn.mssm.edu (N.R.); taylor.ibelli@gmail.com (T.J.I.); akhavan.andre.arya@gmail.com (A.A.A.); jmsharaf24@gmail.com (J.M.S.); peter.henderson@mountsinai.org (P.W.H.); 2SUNY Downstate Medical School, SUNY Downstate Health Sciences University, Brooklyn, NY 11203, USA; christopher.alessandro@downstate.edu; 3The American Medical Program, Tel Aviv University, Tel Aviv 6997801, Israel; rabinovich1@mail.tau.ac.il

**Keywords:** robotic surgery, autologous breast reconstruction, surgical innovation, robotic-assisted surgery, robotic breast surgery, robotic plastic surgery

## Abstract

The expansion of robotic surgery has led to developments in robotic-assisted breast reconstruction techniques. Specifically, robotic flap harvest is being evaluated to help maximize operative reliability and reduce donor site morbidity without compromising flap success. Many publications are feasibility studies or technical descriptions; few cohort analyses exist. This systematic review aims to characterize trends in robotic autologous breast reconstruction and provide a summative analysis of their results. A systematic review was conducted using PubMed, Medline, Scopus, and Web of Science to evaluate robot use in breast reconstruction. Studies dated from 2006 to 2022 were identified and analyzed using the Preferred Reporting Items for Systematic Reviews and Meta-Analyses (PRISMA) guidelines. Full-text, peer-reviewed, English-language, and human subject studies were included. Non-breast reconstruction articles, commentary, expert opinion, editor’s letter, and duplicate studies were excluded. A total of 17 full-text articles were analyzed. The two robotic breast procedures identified were the deep inferior epigastric perforator (DIEP) and the latissimus dorsi (LD) flap. Results showed comparable complication rates and increased operative times compared to NSQIP data on their corresponding open techniques. Additional findings reported in studies included patient reported outcomes, incision lengths, and downward trends in operative time with consecutive procedures. The available data in the literature confirms that robotic surgery is a promising alternative to traditional open methods of breast reconstruction following mastectomy.

## 1. Introduction

Robotic surgery has emerged over the past two decades as an exciting new tool in the surgical armamentarium. Surgical robots have gained traction in numerous fields, including general surgery, orthopedics, gynecology, urology, otolaryngology, and plastic surgery [1,2,3,4,5,6]. The da Vinci^®^ robot was first cleared by the United States Food and Drug Administration (FDA) in 2001 for use in laparoscopic surgical procedures such as removal of the gallbladder and surgery for severe heartburn [7]. Robotic surgery has shown immense promise in reducing adverse outcomes; for example, nerve sparing robotic-assisted prostatectomy has been shown to preserve continence and sexual function [8]. More recently, robotic surgery has been introduced into breast reconstruction, with promising operative and postoperative outcomes.

Despite its growing use, literature on robotic autologous breast reconstruction is limited and heterogeneous. As a newer technology, it is imperative that outcomes and associated costs are critically evaluated. Understanding the intra- and postoperative advantages and disadvantages of robotic breast surgery can better aid surgeons in planning and facilitating breast surgical care and in understanding indications for use of robotics in breast surgery. As technological innovations continue to allow improved operative procedures, it is crucial to assess the advantages and pitfalls of new surgical techniques integrated with technology. Thus, the aim of this systematic review is to characterize the current trends in robotic autologous breast reconstruction (RABR) and provide insight on the current advantages and areas for improvement for each flap described in the literature.

## 2. Materials and Methods

### 2.1. Study Design

A systematic review was conducted using the PubMed, Medline, Scopus, and Web of Science databases to identify RABR articles from January 2006–February 2022, in accordance with the Preferred Reporting Items for Systematic Reviews and Meta-Analyses (PRISMA) guidelines (Figure 1). This systematic review has been registered with PROSPERO (CRD42023425313). Boolean operators were used to identify articles on “Robotic breast surgery”, “Robotic breast reconstruction”, AND “outcome” OR “trend” OR “satisfaction”. Additional searches were performed to identify common RABR procedures, namely “robotic deep inferior epigastric perforator” and “robotic latissimus dorsi flap”. The searches were carried out using the full procedure name, as well as their common abbreviations (“robotic DIEP” and “robotic LD”). No restrictions were used.

Articles were included if the following criteria were met: the full-text article was available, the article was peer-reviewed, all text was written in English, and all subjects were humans who underwent RABR following any type of mastectomy. Non-breast reconstruction-related articles, non-autologous breast reconstruction studies, cadaveric/non-human subject studies, commentary/expert opinion/editor’s letter, review articles, and duplicate studies were excluded.

### 2.2. Data Collection

Two independent reviewers (A.A. and D.R.) screened titles and abstracts for inclusion. Next, four independent reviewers (N.R., T.I., A.A., and D.R.) extracted the following data from the full-text articles: study title, author, year of publication, country of publication, journal of publication, sample size, study aim, patient age, type of robot used, type of flap, immediate vs. delayed reconstruction, reconstruction stage, robotic technique, total operative time, robotic time, and complications (Table 1). Two independent reviewers (A.Y. and P.H.) resolved any conflicts among reviewers.

### 2.3. Data Analysis

The total mean operating time for RABR was calculated by weighing each full-text article by its sample size. Study times were also stratified by immediate or delayed operation types. DIEP and LD procedure times were subsequently compared to flap procedures reported in the National Surgical Quality Improvement Program (NSQIP) database, a nationally validated, risk-adjusted database tracking surgical outcomes [9]. This database acted as the control group and was specifically filtered for DIEP flap breast reconstruction via Current Procedural Terminology (CPT) code 19364 and LD flap reconstruction via CPT code 19361.

**Table 1 jcm-12-04951-t001:** Summary of extracted data across all studies.

**Latissimus Dorsi**												
**Authors (Year)**	**Country**	**Study Design**	**Sample Size**	**Patient Age (Mean ± SD)**	**Type of Robot**	**Immediate or Delayed Reconstruction**	**Laterality**	**Number of Reconstruction Stages**	**Robotic Technique**	**Mean Total Operative Time (Minutes)**	**Mean Robotic Time (Minutes)**	**Complications**
Chung et al. (2015) [10]	South Korea	Retrospective case series	12	35.8 ± 11.8	da Vinci S *	Immediate (n = 4) Delayed (n = 8)	Unilateral	One-stage (n = 4) Two-stage (n = 3) Not applicable (n = 5) **	MP	400.4	85.8	Contour irregularity: 8% (n = 1)
Clemens, M. W., Kronowitz, S., Selber, J. C. (2014) [11]	USA	Retrospective cohort study	12	54.3	da Vinci *	Delayed-Immediate	Unilateral	Two-stage	MP	92	Not specified	Infection: 8% (n = 1) Unplanned OR: 8.3% (n = 1)
Houvenaeghel et al. (2020) [12]	France	Retrospective cohort study	35	51.2 ± 13.5	da Vinci Si: (n = 12) da Vinci Xi: (n = 23) *	Immediate NSM	Unilateral	One-stage	MP-Breast	337	Not specified	Dorsal seroma: 25.7% (n = 9) Hemorrhage/Reoperation: 2.9% (n = 1)
Houvenaeghel et al. (2020) [13]	France	Retrospective comparative study	46	58.1 ± 14.2	da Vinci Si (n = 17) da Vinci Xi (n = 23) *	Immediate SSM	Unilateral	One-stage	MP-Breast	290.5	Not specified	Dorsal seroma: 26% (n = 12) Dorsal bleeding: 2% (n = 1) Infection: 4% (n = 2) Reoperation: 4% (n = 2)
Joo et al. (2021) [14]	South Korea	Case report	1	40	da Vinci SP *	Immediate	Unilateral	One-stage	SP	328	115	None
Lai et al. (2018) [15]	Taiwan	Case report	2	47 ± 0.5	da Vinci Si *	Immediate	Unilateral	One-stage	MP	263	178.5	Seroma: 100% (n = 2)
Moon et al. (2020) [16]	South Korea	Retrospective cohort study	21	29.9 ± 6.8	da Vinci Xi *	Immediate	Unilateral	One-stage	MP-Scapula	295	143	Seroma: 19% (n = 4) Axillary wound complication: 5% (n = 1)
Selber, J. C., Baumann, D. P., Holsinger, C. F. (2012) [17]	USA	Case report	6	Not specified	da Vinci S *	Immediate (n = 3) Delayed (n = 3)	Unilateral	Two-stage	MP	Not specified	111	Transient nerve palsy: 16% (n = 1)
Winocour et al. (2020) [18]	USA	Retrospective review	25	51 ± 9.7	da Vinci *	Not specified	Unilateral	Not specified	MP	366	Not specified	Seroma: 16% (n = 4)
**Deep Inferior Epigastric Perforator**												
**Authors (Year)**	**Country**	**Study Design**	**Sample Size**	**Patient Age (Mean ± SD)**	**Type of Robot**	**Immediate or Delayed Reconstruction**	**Laterality**	**Number of Reconstruction Stages**	**Robotic Technique**	**Mean Total Operative Time (Minutes)**	**Mean Robotic Time (Minutes)**	**Complications**
Bishop et al. (2022) [19]	USA	Retrospective case series	21	54.6 ± 7.6	Not specified	Not specified	Unilateral	Not specified	TAPP	425.3	44.8	Surgical site occurrence (n = 5) Delayed wound healing (n = 1)
Choi et al. (2021) [20]	South Korea	Retrospective case series	17	Not specified	da Vinci SP *	Not specified	Not specified	Not specified	TEP (using single port)	487	65	Not specified
Daar et al. (2022) [21]	USA	Retrospective case series	4	52 ± 6.9	da Vinci Xi *	Both	Bilateral (n = 3) Unilateral (n = 1)	Both	TAPP	717.6	Not specified	Wound dehiscence: 50% (n = 2)
Gundlapalli et al. (2018) [22]	USA	Case report	1	51	da Vinci *	Delayed	Unilateral	Two-stage	TAPP	471	60	None
Kurlander et al. (2021) [23]	USA	Case series	13	50 ± 9.9	Not specified	Immediate	Not specified	One-stage	TAPP	Not specified	Not specified	None
Lee et al. (2022) [24]	South Korea	Retrospective cohort study	21	48.5 ± 6.6	da Vinci SP *	Immediate	Unilateral	One-stage	TAPP	509	Not specified	Flap loss (n = 1), Fat necrosis (n = 2)
Piper et al. (2021) [25]	USA	Case report	4	50.25 ± 8.2	da Vinci Xi *	Immediate	Not specified	One-stage	TAPP	Not specified	45	None
Shakir et al. (2021) [26]	USA	Retrospective cohort study	3	53.2 ± 8.5	da Vinci Xi *	Delayed	Unilateral	Two-stage	TAPP	535	40	None

Addendum: * = Intuitive Surgical, Sunnyvale, CA; ** = Poland Syndrome; MP: Multi-port; NSM: Nipple-sparing mastectomy; SP: Single-port; SSM: Skin-sparing mastectomy; TAPP: Transabdominal pre-peritoneal; TEP: Total extraperitoneal.

Complications from each full-text article were also mapped to relevant NSQIP database complications and to ICD-9 or ICD-10 code sets. Pearson’s chi-square or Fisher exact tests were used to determine associations between surgery type and complications, and odds ratios were then calculated to compare these complication rates.

Statistical analysis was performed by author C.A. using SPSS Statistics for Windows, version 28 (SPSS Inc., Chicago, IL, USA).

### 2.4. Outcomes

The primary outcome of interest was the presence of postoperative complications. Secondary outcomes included operative time, robotic-assisted flap harvest time, robotic technique, and number of reconstruction stages.

## 3. Results

Of the 299 identified articles, a total of 17 full-text articles published from 2006–2022 met inclusion criteria. There were 5 retrospective cohort studies, 5 case reports, 4 retrospective case series, 1 case series, 1 retrospective review, and 1 retrospective comparative study. The mean age of patients was 48.4 years. A total of 84 patients underwent deep inferior epigastric perforator (DIEP) flap breast reconstruction, and 160 patients underwent latissimus dorsi (LD) flap breast reconstruction. The most common type of reconstruction performed was an immediate, one-stage reconstruction. The da Vinci Si robot was the most commonly used device. Other devices used were the da Vinci SP, the da Vinci S, and the da Vinci Xi (Table 1). For all robotic procedures studied, mean total operative time was 394 min and mean robotic time was 88 min. RABR was negatively associated with the need to return to the operating room (OR = 0.149) compared to NSQIP data (Figure 2).

### 3.1. Latissimus Dorsi

Operative techniques used for breast reconstruction with LD were multi-port (MP), MP with breast port access (MP-Breast), MP with scapular port access (MP-Scapula), and single-port (SP) (Figure 3). For patients undergoing LD flap reconstruction, 42 complications were reported (Table 2). Complications ranged from bleeding to seroma formation and nerve palsy. One patient experienced contour irregularity following an LD flap reconstruction. The most common complication among LD flaps was dorsal donor site seroma formation. On average, SP robotic operative time was shortest longest in comparison to other operative techniques, though there was only one case demonstrated (Table 3). Total operative time for LD robotic-assisted procedures was longer (296 min) than NSQIP reported data (256 min) (Figure 4).

### 3.2. Deep Inferior Epigastric Perforator

Operative techniques for the DIEP were transabdominal pre-peritoneal (TAPP) vs. total extraperitoneal (TEP) approaches for the DIEP (Figure 3). The TAPP procedures were performed with a multi-port device. The TEP procedure was completed with a single-port (da Vinci SP). For patients undergoing DIEP flap reconstruction, 10 complications were reported (Table 2). The most common specified complication among DIEP flaps was wound dehiscence (Table 2). On average, TAPP robotic operative time was shorter in comparison to TEP (Table 3). Total operative time for DIEP robotic-assisted procedures was longer (546 min) than NSQIP reported data (488 min) (Figure 4).

## 4. Discussion

Breast reconstruction rates continue to rise, highlighting the importance of diverse treatment options and favorable outcomes. Numerous factors, such as age, socioeconomic status, geographic location, ethnicity, and patient preference, influence the decision to undergo breast reconstruction following mastectomy [27]. Reconstruction has been linked to lower anxiety and depression levels and improved quality of life compared to mastectomy alone [28,29].

In recent years, robotic mastectomy has emerged as an innovative technique in breast cancer surgery, offering potential advantages such as reduced blood loss, smaller incisions, faster recovery times, and improved cosmetic outcomes [30]. As the adoption of robotic mastectomy increases, RABR may serve to further enhance patient outcomes and satisfaction.

Robotic reconstruction, which offers potential advantages such as tremor elimination, increased surgical dexterity, and minimally invasive approaches, is a promising technique to address the ever-present demand for improved reconstructive outcomes. Further research and technological advancements may help to make robotic reconstruction more accessible and appealing, ultimately decreasing operative morbidity and improving patients’ psychological well-being and quality of life after mastectomy.

Comparing operative times, complication rates, patient-reported outcome measures (PROMs), and fascial incision data from robotic and traditional reconstructive surgery literature can provide valuable insights into the feasibility of robotic reconstruction techniques.

### 4.1. Operative Time

It is not surprising that robotic procedures take longer compared to their open counterparts due to the need for device setup and troubleshooting. In a comparison of operative times for deep inferior epigastric perforator (DIEP) flap breast reconstruction, robotic-assisted procedures averaged 546 min, while data from the National Surgical Quality Improvement Program (NSQIP) reported an average of 488 min for non-robotic procedures. Issa et al. found mean operative times of 506.3 min for immediate bilateral DIEP flap reconstruction using an open approach and 464.8 min for delayed bilateral DIEP flap reconstruction [31]. Similarly, the robotic LD averaged a longer operative time than open (296 and 256 min, respectively). Despite the slightly longer duration for robotic-assisted procedures, the difference in operative time remains relatively modest when compared to non-robotic methods. While the robotic method certainly requires more training and longer operative time initially as an investment, increased experience with robotic surgery may lead to improved operative times and efficiency, as suggested by Moon et al., who reported a downward trend in operative times with each subsequent case performed [32].

### 4.2. Complication Rates

Complication rates between RABR and traditional techniques appear to be comparable, as shown in Figure 2. Due to the nature of our study, direct comparison with NSQIP data is limited; however, existing literature supports similar findings. Clemens et al. reported a 16.7% overall complication rate for robotic assisted LD versus 37.5% for traditional open latissimus dorsi (LD) reconstruction [11], while Houvenaeghel et al. found rates of 28.6% for robotic assisted LD compared to 51.4% for the traditional LD approach [12]. Notably, Bishop et al. observed decreased pain on the robotic side for patients who underwent bilateral DIEP surgery with one robotic and one open procedure [19]. While the robotic re-operation rate in our study was promising and proved to be statistically lower than the NSQIP data, the robotic procedures are new and the follow-up time is relatively short, so may not account for re-operation events further in the future.

### 4.3. Patient-Reported Outcome Measures

Several studies have investigated patient-reported outcome measures (PROMs) for robotic-assisted breast reconstruction (RABR). Chung et al. found high mean satisfaction scores across various categories, such as general satisfaction, scar formation, and breast aesthetics [10]. Joo et al. used the BREAST-Q tool and reported higher mean overall scores for RABR patients compared to those who underwent traditional breast reconstruction [14]. Similarly, Moon et al. observed favorable mean PROMs for improvement in chest deformity, chest symmetry, scar formation, and overall satisfaction [16]. These findings suggest that RABR is associated with positive patient-reported outcomes; however, a prospective study directly comparing BREAST-Q outcomes of robotic and traditional techniques has not been conducted.

### 4.4. Incision Length

The minimally invasive nature of robotic surgery results in shorter scars in latissimus-based breast reconstruction [14] and limited fascial incisions in abdominally based breast reconstruction. Several studies have reported fascial incision lengths for DIEP flaps, highlighting the potential benefits of a smaller incision. While long-term data is limited, a reduced fascial incision length may decrease the risks of hernia and bulge, which are common and potentially underreported complications of DIEP flaps. Reported robotic fascial incision lengths ranged from 1.5 cm to 7 cm. Kurlander et al. discussed using preoperative computed tomography angiography (CTA) to determine robotic DIEP candidacy and reported a mean fascial incision length of 3.5 cm in their preselected population. This is markedly shorter than the mean fascial length of a traditional open DIEP flap of 13.3 cm [33]. Given the potential postoperative benefits of reduced long-term risk for abdominal bulge or hernia, plastic surgeons should consider this metric when evaluating the robotic approach.

Although robotic-assisted breast reconstruction has many apparent benefits, potential drawbacks of robotic-assisted surgery must also be considered. These include longer operative times, variable postoperative outcomes, and increased financial burden [34,35,36]. The time-consuming start-up, docking, and setup of the robot, as well as its occupation of operating room space, may impact operative efficiency. The choice between a single-port or multi-port robotic system can also affect the docking process, camera flexibility, and potential for robotic arm collisions [14]. The controversy surrounding robotic surgery stems from its challenge to the status quo of reconstructive procedures and the financial and technical obstacles it presents. More research is needed to form definitive conclusions, such as the number of procedures required to optimize operative time when using a robotic approach, the optimal device and technique to use for each procedure, and the relative risk/benefit ratio of outcomes such as bleeding and hernia.

To our knowledge, this represents the first systematic review that compares complication types and rates for robotic-assisted breast reconstruction (RABR) with nationally collected data on surgical outcomes in traditional open flap autologous breast reconstruction. We further characterized RABR by surgical technique and compared operative times among procedures. Previous systematic reviews have also assessed robotic techniques in autologous breast reconstruction. Donnelly et al. conducted a systematic review with findings comparable to ours, focusing on financial cost and the total length of the hospital stay as key outcomes [37]. Our study builds upon their work by incorporating data from Medline, PubMed, and Web of Science to capture a larger pool of published studies.

Dobbs et al. conducted a comprehensive, systematic review in 2017, revealing a steady increase in publications on robotic surgery in general over the past 20 years, from 168 in 2000 to over 2000 in 2014 [6]. There seems to be a similarly growing interest in robotic surgery among plastic surgeons in publications and conferences, emphasizing the need for further evaluation and in-depth study of its benefits and drawbacks.

This study rigorously characterizes the role of robotic surgery in autologous breast reconstruction, from its first published article in 2004 to the present day. However, it has limitations and challenges. Primarily, our findings are based on case reports, case series, and non-randomized cohort studies, which may contain biases. Consequently, conclusions should be approached with caution.

Over half of the included studies did not differentiate between total operative and robotic time, making it difficult to determine if the robotic component influences operative time or financial costs. Furthermore, studies lacked consistent reporting of patient demographics and comorbidities, which are crucial factors in postoperative complications. This makes it challenging to ascertain if robotic surgeries or specific flap types lead to a higher incidence of complications compared to traditional techniques.

Direct comparison with NSQIP data also has its limitations, as the CPT code for breast reconstruction with free flap (19364) does not distinguish between abdominal donor site (DIEP) versus other types of flaps such as thigh or gluteal. However, we expect the data to trend toward abdominal flaps as they are much more commonly performed than other flaps for breast reconstruction [38]. The NSQIP database also does not specify between unilateral and bilateral DIEP flap reconstruction. As such, we were unable to draw a direct comparison between unilateral and bilateral operative times for NSQIP-reported data and the data obtained from this systematic review. Lastly, as robotic technology is still emerging in plastic surgery, these preliminary results may not fully represent the potential of RABR.

As the field of RABR is still developing, future studies should focus on conducting prospective cohort studies in eligible patients to compare the efficacy of robotic and traditional surgical techniques across various free and pedicled flaps. Additionally, it is crucial for studies to emphasize the significance of patient selection when reporting results, allowing for a better understanding of the ideal candidates for RABR and facilitating more tailored approaches in breast reconstruction.

The widely used robotic systems in surgical procedures, including the RABR procedures included in this analysis, were not designed with reconstructive surgery in mind. Although the system has demonstrated potential benefits, it is crucial to acknowledge its limitations in the context of breast reconstruction. These limitations include high costs, a steep learning curve, ergonomic considerations, limited haptic feedback, bulky equipment, and a lack of dedicated instruments tailored for plastic surgery [39]. Despite these challenges, advancements in technology hold promise for the development of future robotic systems that can overcome these limitations. Improved haptic feedback, miniaturization, dedicated instruments, cost reduction, and integration with imaging technologies may enhance the effectiveness and versatility of robotic assistance in plastic surgery. The Symani Surgical System^®^ (Medical Microinstruments, MMI, Calci, Italy) is a relatively novel robotic microsurgical system that consists of enhanced magnification and flexible robotic arms that can reach into deeper anatomical regions [40]. It has been utilized for numerous plastic surgery procedures, including lymphedema and autologous breast reconstruction via the profunda artery perforator flap [40]. Surgeons continue to engage with new robotic surgical systems in reconstructive surgery and microsurgery, contributing to innovation and advancement in the field of plastic surgery.

## 5. Conclusions

Operative efficiency, postoperative clinical outcomes, and patient satisfaction are all of great importance to the plastic surgeon. Robotic-assisted breast reconstruction has been shown to be a feasible procedure with advantages including smaller skin or fascial incision sizes and reduced complication rates. Smaller fascial incision sizes may lead to decreased long-term postoperative complications, such as abdominal hernia and chronic postoperative pain, though these complications can take many years to present. There is still limited data in this field regarding operative efficiency, cost, and maximal patient benefit. A continued effort to understand how robotic assistance can aid the plastic surgeon will hopefully shed more light on the robot as a tool of the future for breast reconstruction. 

## Figures and Tables

**Figure 1 jcm-12-04951-f001:**
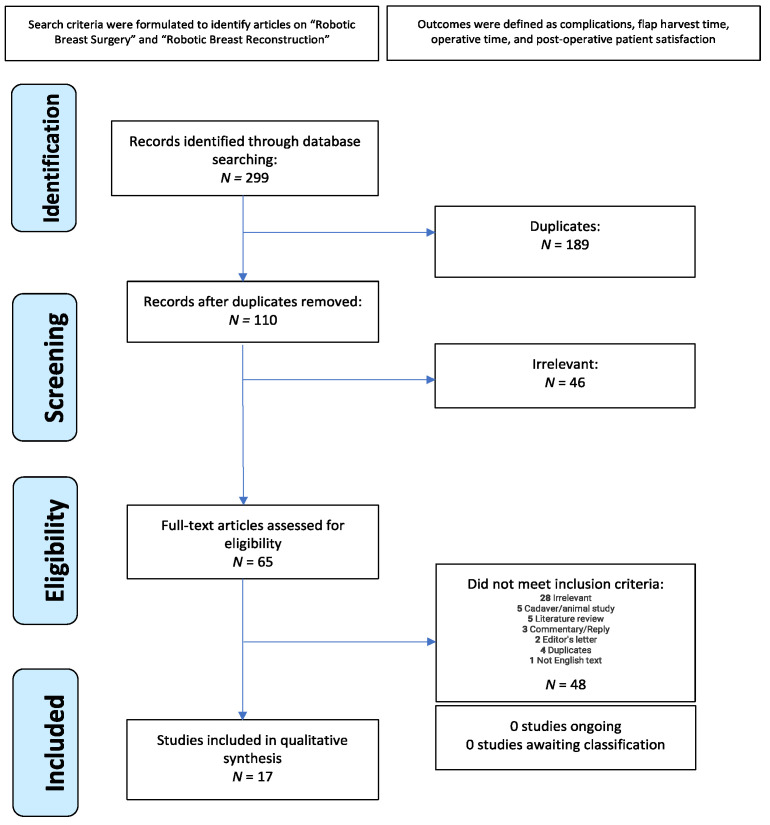
Preferred reporting items for systematic reviews and meta-analyses (PRISMA) protocol.

**Figure 2 jcm-12-04951-f002:**
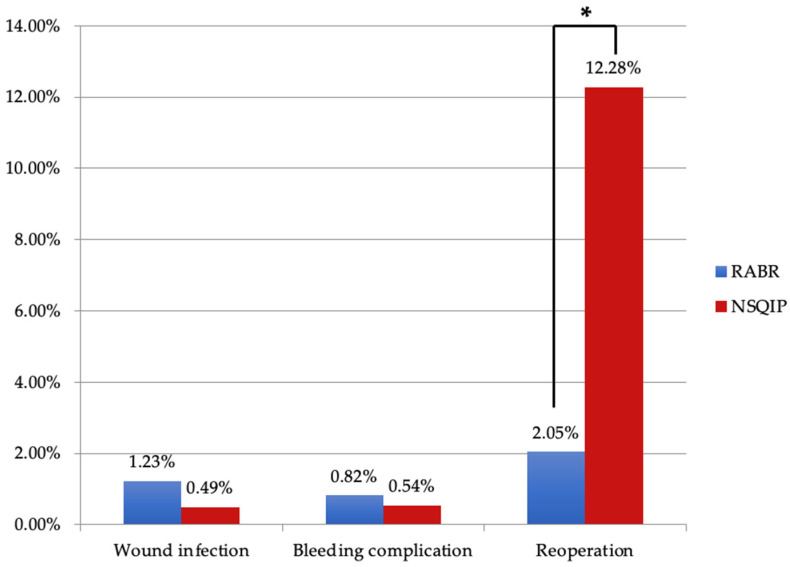
Comparison between RABR and NSQIP complication profiles. Key: RABR: Robotic Autologous Breast Reconstruction; NSQIP: National Surgical Quality Improvement Program. *—indicates significance, *p* < 0.05.

**Figure 3 jcm-12-04951-f003:**
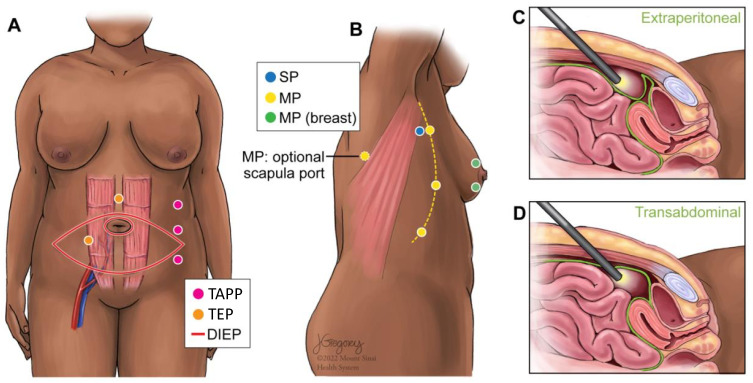
Schematic of port access sites for latissimus dorsi and DIEP flap robotic-assisted surgery. (**A**) Port site variations for the robotic DIEP. (**B**) Port site variations for the robotic LD. (**C**) Depth of port for DIEP-TEP. (**D**) Depth of port for DIEP-TAPP. Key: DIEP: Deep inferior epigastric perforator; MP: Multi-port; SP: Single-port; TAPP: Transabdominal pre-peritoneal; TEP: Total extraperitoneal.

**Figure 4 jcm-12-04951-f004:**
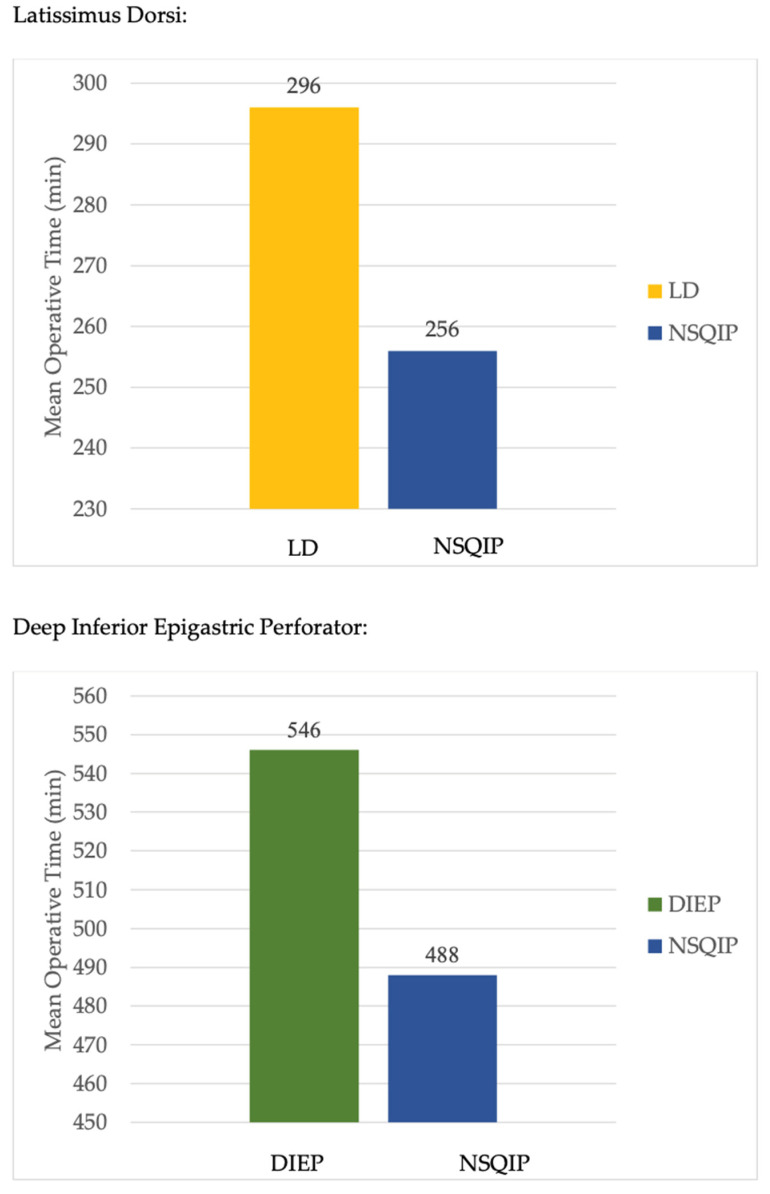
Mean operative times by reconstruction type. Key: DIEP: Deep inferior epigastric perforator; LD: Latissimus dorsi; NSQIP: National Surgical Quality Improvement Program.

**Table 2 jcm-12-04951-t002:** Complications by flap type.

**Latissimus Dorsi**			
**Type of Flap**	**Complication**	**Number of Patients (n)**	**Percentage (%)**
**LD** (n = 160)	Donor site seroma	31	19.3%
	Re-operation	3	1.8%
	Contour irregularity	1	<1%
	Infection	3	1.8%
	Hemorrhage requiring reoperation	1	<1%
	Dorsal bleeding	1	<1%
	Other wound complication (axillary)	1	<1%
	Nerve palsy	1	<1%
	**Total**	42	19%
**Deep Inferior Epigastric Perforator**			
**Type of Flap**	**Complication**	**Number of Patients (n)**	**Percentage (%)**
**DIEP** (n = 84)	Wound dehiscence	3	3.6%
	Flap loss	1	1.2%
	Fat necrosis	2	2.4%
	Other (not specified)	4	4.8%
	**Total**	10	12%

Key: DIEP: Deep inferior epigastric perforator; LD: Latissimus dorsi.

**Table 3 jcm-12-04951-t003:** Patient distribution and mean operative time by surgical approach.

**Mean Operative Time: Latissimus Dorsi**				
**Robotic Technique**	**Number of Procedures**	**Mean Operative Time (minutes)**		**Ratio Robotic to Total Operative Time**
		Total	Robotic	
MP	57	280	125	44%
MP-Breast	81	314	−	−
MP-Scapula	21	295	143	48%
SP	1	328	115	35%
**Mean Operative Time: Deep Inferior Epigastric Perforator**				
**Robotic Technique**	**Number of Procedures**	**Mean Operative Time (minutes)**		**Ratio Robotic to Total Operative Time**
		Total	Robotic	
TAPP	67	531	47	8%
TEP	17	487	65	13%

Key: MP: Multi-port; SP: Single-port; TAPP: Transabdominal pre-peritoneal; TEP: Total extraperitoneal.

## Data Availability

No new data were created or analyzed in this study. Data sharing is not applicable to this article. No review protocol was prepared for this study.

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
