# Peer review of "The Expanding Utility of Robotic-Assisted Flap Harvest in Autologous Breast Reconstruction: A Systematic Review"

_jcm, 2023, doi:10.3390/jcm12154951_

Round 1

Reviewer 1 Report

Thank you very much for the opportunity to read this manuscript. This paper is very interesting and takes up an interesting systemic review. The constantly evolving science and technology used in surgery can significantly improve therapeutic outcomes in skin flaps.

My concerns : 

1) Please spell out all abbreviations under the tables. Most are there but e.g. in Figure 4 - NSQIP etc.

2) There is no need to write FEBOPRAS etc next to the names. please adjust to MDPI standard

3) Prisma Flow Chart has been provided, but I think it would be good to add PRISMA Checklist to Supplementary Material. 

4) Although the paper deals with robotic procedures, I found an interesting paper on flap survival using PRP treatment - maybe it is worth confronting this element in the discussion as well - on the one hand, the use of robots may make the flap preparation procedure more accurate and efficient. 

5) The conclusions are too general for me, that what matters for plastic surgeons is a small scar and thus patient satisfaction.  Rather, the results of this analysis should lead to a reflection in the conclusions (why should we, for strictly medical reasons, use this type of device like the Da Vinci) a lower rate of vascular complications ?  Lower rate of flap necrosis ? 

Therefore, the authors should consider extending the discussion to include possible complications with this type of procedure such as necrosis of the flaps.  PMID : 36983115 - where they extensively present the effect of PRP injection on improving the blood supply to the flaps. Perhaps a similar study on DIEP ? I would like to consider this suggestion.

With best regards,

Reviewer 2 Report

This study made a review of RABR researches. The authors disclosed the advantages and limitation of this robotic aid surgery. That is quite impressive. Here is a question if it is possible to analyze the difference of using the different surgical robot in RABR. Maybe some of them can profit in the cost, learning curve, etc.  

Round 2

Reviewer 1 Report

Accept as it is.

Authors well adressed most of suggestions.